# Organelle Comparative Genome Analysis Reveals Novel Alloplasmic Male Sterility with *orf112* in *Brassica oleracea* L.

**DOI:** 10.3390/ijms222413230

**Published:** 2021-12-08

**Authors:** Li Chen, Wenjing Ren, Bin Zhang, Wendi Chen, Zhiyuan Fang, Limei Yang, Mu Zhuang, Honghao Lv, Yong Wang, Jialei Ji, Yangyong Zhang

**Affiliations:** 1Key Laboratory of Biology and Genetic Improvement of Horticultural Crops, Institute of Vegetables and Flowers, Chinese Academy of Agricultural Sciences, Ministry of Agriculture, Beijing 100081, China; 18205480752@163.com (L.C.); 17863805323@163.com (W.R.); 13126720352@163.com (B.Z.); chenwd9611@163.com (W.C.); fangzhiyuan@caas.cn (Z.F.); yanglimei@caas.cn (L.Y.); zhuangmu@caas.cn (M.Z.); lvhonghao@caas.cn (H.L.); wangyong@caas.cn (Y.W.); jijialei@caas.cn (J.J.); 2State Key Laboratory of Crop Genetics and Germplasm Enhancement, College of Horticulture, Nanjing Agricultural University, Nanjing 210095, China

**Keywords:** cabbage, organelle genome, alloplasmic male sterility, comparative genome, Bel CMS

## Abstract

*B. oleracea* Ogura CMS is an alloplasmic male-sterile line introduced from radish by interspecific hybridization and protoplast fusion. The introduction of alien cytoplasm resulted in many undesirable traits, which affected the yield of hybrids. Therefore, it is necessary to identify the composition and reduce the content of alien cytoplasm in *B. oleracea* Ogura CMS. In the present study, we sequenced, assembled, and compared the organelle genomes of Ogura CMS cabbage and its maintainer line. The chloroplast genome of Ogura-type cabbage was completely derived from normal-type cabbage, whereas the mitochondrial genome was recombined from normal-type cabbage and Ogura-type radish. Nine unique regions derived from radish were identified in the mitochondrial genome of Ogura-type cabbage, and the total length of these nine regions was 35,618 bp, accounting for 13.84% of the mitochondrial genome. Using 32 alloplasmic markers designed according to the sequences of these nine regions, one novel sterile source with less alien cytoplasm was discovered among 305 materials and named Bel CMS. The size of the alien cytoplasm in Bel CMS was 21,587 bp, accounting for 8.93% of its mtDNA, which was much less than that in Ogura CMS. Most importantly, the sterility gene *orf138* was replaced by *orf112*, which had a 78-bp deletion, in Bel CMS. Interestingly, Bel CMS cabbage also maintained 100% sterility, although *orf112* had 26 fewer amino acids than *orf138*. Field phenotypic observation showed that Bel CMS was an excellent sterile source with stable 100% sterility and no withered buds at the early flowering stage, which could replace Ogura CMS in cabbage heterosis utilization.

## 1. Introduction

Cabbage (*Brassica oleracea* L. var. *capitata*) is an important leafy vegetable crop that is widely cultivated worldwide. As a typical cross-pollinated plant, cabbage exhibits noticeable heterosis, and hybrids usually have better growth potential, yield, and disease resistance than their parents [1]. The utilization of heterosis in Brassica crops depends mainly on either self-incompatible or male-sterile lines [2].

Cytoplasmic male sterility (CMS) is maternally inherited. With CMS, the pistil develops normally, but the stamen is abnormal and cannot produce functional pollen [3,4]. The use of CMS not only ensures the quality and purity of hybrids, but also protects the intellectual property rights related to the parents. Therefore, cytoplasmic male-sterile lines have gradually replaced the use of self-incompatible lines as the main method of hybrid production in many *Brassica* crops [2]. CMS occurs widely among higher plants, and CMS caused by natural variation in the mitochondrial genome is usually referred to as homoplasmic CMS. Examples of homoplasmic CMS include CMS-T in maize [5], Pol CMS and Nap CMS in rapeseed [6,7], and NWB CMS and DCGMS in radish [8,9]. To date, no occurrences of homoplasmic CMS have been found in *B. oleracea* crops. CMS derived from interspecific or intergeneric hybridization and somatic cell fusion is often referred to as alloplasmic CMS [10]. The most widely used alloplasmic male-sterile line in *B. oleracea* is the Ogura CMS, which was introduced from radish [1]. Although the Ogura CMS line has greatly improved the seed purity of *B. oleracea* and has undergone several rounds of improvements [11,12], many undesirable traits affecting seed yield are still present, such as a high rate of withered buds, small nectaries, a light petal color, and low seed production. Previous studies in Solanaceae and Cruciaceae crops have shown that alloplasmic CMS retains many redundant donor mitochondrial genome sequences [13,14,15,16]. It is possible that the exogenous mitochondrial fragment could not coordinate with the nuclear genome and resulted in those undesirable traits. Identifying the composition and content of the exogenous fragments is of great importance to solve the problem.

Chloroplasts and mitochondria are essential semiautonomous organelles for higher plants [17]. Many studies have shown that chloroplasts and mitochondria are not only responsible for photosynthesis and respiration, but also associated with many various traits, such as seed oil content, cold tolerance, and sex differentiation [18,19,20]. Therefore, the chloroplast and mitochondrial genomes are important molecular bases for higher plant cell functions. The chloroplast genomes (cpDNAs) of higher plants are structurally simple, small (100~300 kb), and stably inherited with nearly no recombination [21]. In contrast, plant mitochondrial genomes (mtDNAs) are highly variable in terms of size and structure [22]. For example, the mtDNA in *Brassica* is approximately 200 kb [23], but the mtDNA in *Silene conica* is approximately 11.3 Mb [24]. Interestingly, there is no linear relationship between the size of mtDNA and the number of encoded genes. In addition, plant mitochondrial genomes have complex structures, which are mainly the result of frequent recombination events and gene rearrangement [25,26]. Abnormal recombination and nonhomologous end-joining (NHEJ) activities often lead to the formation of novel open reading frames (ORFs) [27]. These novel genes could possibly form chimeric structures with known mitochondrial genes and encode proteins with transmembrane domains, which may cause abnormal development of flower organs or pollen grains and eventually result in CMS [28].

In this study, the complete chloroplast and mitochondrial genomes of Ogura CMS cabbage and its maintainer line were obtained via second- and third-generation sequencing technologies. The size and composition of alien fragments in the Ogura-type cabbage mitochondrial genome were determined by comparative genome analysis. Based on these results, a set of alloplasmic screening markers was developed, and a novel excellent sterility source, Bel CMS, was identified using this screening system, which had less alien cytoplasm and no withered buds at early flowering stage. These results lay the foundation for heterosis utilization and cytoplasmic improvement in *B. oleracea*.

## 2. Results

### 2.1. Analysis of Ogura- and Normal-Type Chloroplast Genomes

Using second-generation sequencing data, we assembled the chloroplast genomes of normal-type and Ogura-type cabbage into single circular molecules with sizes of 153,365 bp and 153,363 bp, respectively (Figure 1A). The overall GC content of both cpDNAs was 36.36%, which was comparable to that of other cpDNAs among Brassicaceae. The two chloroplast genomes had similar structures, including a long single copy (LSC), a short single copy (SSC), and a pair of inverted repeats (IRA and IRB) (Figure 1B).

In total, 131 genes were identified in the chloroplast genome, consisting of 86 protein-coding genes, eight rRNA genes, and 37 tRNA genes. The protein-coding genes mainly included ATP synthase, the cytochrome b/f complex, NADH dehydrogenase, photosystem, ribosomal proteins, and other related genes (Table 1). Among the 131 genes, 114 genes had no introns, 15 genes (*trnK*-UUU, *rps16*, *trnG*-UCC, *atpF*, *rpoC1*, *rnL*-UAA, *trnV*-UAC, *petB*, *petD*, *rpl16*, *rpl2*, *ndhB*, *trnI*-GAU, *trnA*-UGC and *ndhA*) contained one intron, two genes (*clpP* and *ycf*) contained two introns, and one gene (*rps12*) exhibited trans-splicing. Compared with Ogura-type radish (NC_024469.1), the chloroplast genome of cybrid 19-3-2 was identical to normal-type cabbage in structure and coding sequence, indicating that the chloroplast genome of cybrid 19-3-2 was completely derived from *B. oleracea* (Appendix A).

### 2.2. Analysis of Ogura- and Normal-Type Mitochondrial Genome

Because the mitochondrial genomes of higher plants usually exhibit considerable structural variation, we used second- and third-generation sequencing technologies to assemble the mitochondrial genomes to ensure accuracy. The mitochondrial genome of normal-type cabbage 19-3-1 was ultimately assembled into a single circular molecule of 219,969 bp with a GC content of 45.25% (Figure 2A). However, the mitochondrial genome of Ogura-type cabbage 19-3-2 was assembled into two circular molecules of different sizes (Figure 2B). The small circle (19-3-2_MT1) comprised 71,998 bp with a GC content of 45.56%, and the large circle (19-3-2_MT2) comprised 185,431 bp with a GC content of 45.25%.

The normal- and Ogura-type mitochondrial genomes were annotated via BLAST homology alignment and tRNAscan-SE. The coding genes within the two types of mitochondrial genomes were relatively conserved. A total of 33 protein-coding genes, three rRNA genes, and 18 tRNA genes were detected in the normal-type mitochondrial genome. The 33 protein-coding genes included nine subunits of complex I (*nad1*, *2*, *3*, *4*, *4 L*, *5*, *6*, *7*, and *9*), one subunit of complex III (*cob*), four subunits of complex IV (*cox1*, *2-1*, *2-2* and *3*), five subunits of complex V (*atp1*, *4*, *6*, *8*, and *9*), five cytochrome c biogenesis genes (*ccmB*, *ccmC*, *ccmFN1*, *ccmFN2* and *ccmFC*), eight ribosomal genes (*rpl2*, *rpl5*, *rpl16*, *rps3*, *rps4*, *rps7*, *rps12* and *rps14*), and one maturase gene (*matR*). The mitochondrial genome of Ogura-type cabbage contains all of the above genes as well as an additional sterility gene *orf138* and three tRNA genes (*trnfM*-CAU, *trnN*-GUU and *trnY*-GUA) (Appendix A).

### 2.3. Synteny Analysis of the Mitochondrial Genomes

To further clarify the structure and composition of the Ogura-type cabbage mitochondrial genome, syntenic analysis was performed with the normal-type cabbage and the Ogura-type radish mitochondrial genomes (NC_018551) (Figure 3A,B). The results showed that there were 13 structural variations (>1 kb in size) between Ogura- and normal-type cabbage mtDNA, while there were 22 structural variations (>1 kb in size) between Ogura-type cabbage and radish mtDNA. Moreover, the structural variations between Ogura-type cabbage and radish were more complex, and the syntenic regions were short and scattered. Therefore, the mtDNA of Ogura-type cabbage exhibited higher synteny with normal-type cabbage than with Ogura-type radish. We designed 15 pairs of primers based on the structural variations between Ogura- and normal-type cabbage mtDNA (Appendix A). The PCR results showed that all the primers had amplification products in Ogura-type cabbage, while no amplification was detected in normal-type cabbage (Figure 3C), which further confirmed the structural variations between the Ogura- and normal-type cabbage mtDNA. We also compared the coding sequences of the three mitochondrial genomes. With the exception of a SNP (1398 A/G) within the *rrn18* gene, the coding sequences of Ogura- and normal-type cabbage mtDNA were consistent. However, compared to Ogura-type radish (NC_018551), there were 16 genes with at least one base mutation in Ogura-type cabbage, some of which resulted in amino acid changes (Table 2).

### 2.4. Development and Application of Alloplasmic Markers

Sequence alignment showed that nine specific regions in the Ogura-type cabbage mitochondrial genome were completely derived from Ogura-type radish, and among these regions, the Ogura sterility gene *orf138* was located in region R8 (Figure 4A). The nine specific regions varied in size, with the smallest R1 (1551 bp) and the largest R9 (6820 bp) (Table 3). The total length of the nine specific regions was 35,618 bp, accounting for 13.84% of the Ogura-type cabbage mtDNA. Moreover, the gene annotation results showed that the nine specific regions did not encode any known mitochondrial genes, except for the sterility gene *orf138*.

Based on the sequences of the nine specific regions, a total of 32 specific markers were developed to identify the alien cytoplasm (Appendix A). These markers were used to identify the cytoplasm of 305 cabbage materials. The detection results of 152 materials were consistent with those of normal-type cabbage, indicating that these materials did not have alien cytoplasm and were not from the Ogura CMS. The other 150 materials were consistent with Ogura-type cabbage, indicating that they had the same alien cytoplasm as the Ogura-type cabbage. Notably, three materials, namely 19-2202, JY314, and 18Q2183, were significantly different from both Ogura-type and normal-type cabbage. Only 16 out of the 32 markers were detected in these three materials, and the markers representing the specific regions of R2 to R6 were not amplified (Figure 4B), which indicated that the content of alien cytoplasm in these three materials was much lower than that in the Ogura-type cabbage. In addition, we found that the amplification product of the Ogura sterility gene *orf138* in these three materials was slightly shorter than that in Ogura-type cabbage. Sequencing and comparison results showed that the *orf138* gene in these three materials had a 78-base deletion. However, there were no SNPs, and the amino acid sequence on either side of the deletion was not changed. Because the new gene encoded 112 amino acids, we named it *orf112*. Based on the differences in the sterility genes and alien cytoplasm composition, we considered this material to be a new type of CMS line distinct from Ogura CMS and named it Bel CMS. We then sequenced the chloroplast and mitochondrial genomes of the Bel CMS to determine the exact composition of the radish fragment in its mitochondrial genome.

### 2.5. Analysis of the Chloroplast and Mitochondrial Genomes of Bel CMS

The chloroplast genome of Bel CMS was ultimately assembled into a single circular molecule comprised of 153,377 bp and a CG content of 36.37%. Annotation and comparison showed that the coding genes and structure of Bel CMS were the same as those of Ogura- and normal-type cabbage (Appendix A).

Similar to the Ogura-type cabbage mitochondrial genome, the Bel CMS mitochondrial genome was also assembled into two circles (Figure 5). The small circle (19-2202_MT1) comprised 65,849 bp with a CG content of 45.57%. The large circle (19-2202_MT2) comprised 175,791 bp with a CG content of 45.04%. The gene annotation results showed that the Bel CMS mitochondrial genome encoded a total of 57 known genes, including 34 protein-coding genes, three rRNA genes, and 20 tRNA genes (Appendix A). Except for the sterility gene *orf112*, the other protein-coding genes did not differ from those in Ogura-type cabbage.

### 2.6. Structural Variations and Alloplasmic Composition of the Bel CMS Mitochondrial Genome

Synteny analysis showed that there were 10 structural variations (>1 kb in size) between the Bel CMS mitochondrial genome and the normal cabbage mitochondrial genome (Figure 6A). In addition, there were 21 structural variations (>1 kb in size) between the Bel CMS mitochondrial genome and the radish mitochondrial genome (Figure 6B). Compared to Ogura-type cabbage, Bel CMS exhibited a higher synteny with normal cabbage, which indicated that the mitochondrial genome of Bel CMS was more closely related to that of normal cabbage in terms of genetic relationships.

The sequencing and comparison results showed that a total of four specific regions in the Bel CMS mitochondrial genome were completely derived from Ogura-type radish (Figure 6C, Table 4). The first three specific regions in Bel CMS corresponded to R7, R1, R8 and R9 in Ogura-type cabbage, which was consistent with our detection results using alloplasmic markers (Figure 4B). The fourth region, R10, was not present in Ogura-type cabbage and was unique to 19-2202. The total length of alien cytoplasm in the Bel CMS was 21,587 bp, accounting for 8.93% of the whole mtDNA, which was much lower than the proportion of alien cytoplasm in Ogura-type cabbage (13.84%).

### 2.7. Comparison of Phenotypes of Ogura CMS and Bel CMS Cabbage under Field Conditions

To compare the performance of Bel CMS and Ogura CMS in terms of various breeding traits, 19-2202 and 19-3-1 were used as the male parent and female parent, respectively, for crossing and successive backcrossing. Phenotypic observations of thirty F_1_-generation plants revealed no significant differences in flower organ development (Figure 7C) or degree of sterility (Figure 7D–F) between Bel CMS and Ogura CMS under the same field conditions. The main difference was that Bel CMS had no withered buds at the early flowering stage, while Ogura CMS displayed severe bud mortality (Figure 7A,B). Thus, the discovery of Bel CMS is of great significance for the heterosis utilization in *B. oleracea*.

## 3. Discussion

### 3.1. Sequencing and Assembly of Plant Organelle Genomes

With the development of sequencing technology, the complete sequences of many plant organelle genomes have been released. To date, chloroplast and mitochondrial genome sequencing has been completed for 6284 and 419 plant species, respectively (https://www.ncbi.nlm.nih.gov/genome/browse#!/organelles (accessed on 2 December 2021)). Chloroplast genomes are easy to assemble due to their simple structure and low degree of variation [21]. However, plant mitochondrial genomes contain many repeat sequences as well as inserts of nuclear and chloroplast origin, making their assembly difficult [10]. Because third-generation sequencing technology can accurately identify structural variation, it has advantages for the sequencing and assembly of mitochondrial genomes with complex structures. Further, the second generation sequencing technology has high sequencing accuracy. The difficulty of assembling plant mitochondrial genomes is greatly reduced through second- and third-generation sequencing technology, which greatly promotes the composition analysis of plant mitochondrial genomes [29,30,31].

In this study, the complete cabbage chloroplast genome sequences of one fertile line and two male-sterile lines were obtained using Illumina HiSeq data. The three chloroplast genomes consisted of an LSC, an SSC, and a pair of inverted repeats (IRA and IRB) (Figure 1 and Appendix A), which are typical structures of plant chloroplast genomes, and the coding genes were highly conserved among these three genomes. This result is consistent with previous results. Kim et al. (2018) sequenced and compared the chloroplast genomes of 28 *Brassica* crop plants and found that the chloroplast genomes of cabbage plants with the C genome present low intraspecies diversity compared to those of other *Brassica* crop species [32].

Using Illumina HiSeq data and Nanopore Sequel data, we obtained three cabbage mitochondrial genomes. The genome of the fertile line was a single circular molecule (Figure 2A), but the genomes of other two male-sterile lines contained two circular molecules (Figure 2B and Figure 5), which was different from some other findings in cruciferous crops [4,33,34]. In fact, plant mitochondrial genomes can have various forms, such as linear forms [35,36], mixed linear and circular forms [37,38], single circular molecules [10,33,39], and one main circular molecule plus several subgenomes [40]. With the development and application of third-generation sequencing technology, an increasing number of plant mitochondrial genomes with polycyclic structures have been identified, especially in cytoplasmic male-sterile lines. The mitochondrial genome of the commercial sugarcane species Khon Kaen 3 was assembled into two circles of different sizes via PacBio Sequel [41]. Similarly, using specific length readings generated by PacBio Sequel, it was demonstrated that three isoforms of mtDNA molecules exist in the radish NWB CMS mitochondrial genome [4]. The recombination of repeated sequences leads to different isoforms of mitochondrial genome [42], and the variation of alloplasmic CMS mitogenome is likely to result from protoplast fusion and interspecific hybridization.

### 3.2. The Alien Cytoplasm MAY Affect the Agronomic Traits of Alloplasmic CMS Lines

In sexual reproductive plants, organelle genomes follow the principle of maternal inheritance, but in somatic hybridization, organelle genomes are inherited in a more complex manner [43]. It has been shown that chloroplasts generally do not fuse with each other during plant growth and development; that is, only one parent chloroplast is present in somatic hybrids. In contrast, mitochondria usually undergo fusion, which means that the mitochondria of somatic hybrids contain exogenous fragments from the donor parent, especially in asymmetric fusion [13,44,45]. In this study, the chloroplast genomes of the two alloplasmic cytoplasmic male-sterile lines examined were derived completely from cabbage, while the mitochondrial genomes consisted of cabbage components as well as components of radish cytoplasm. These results were consistent with those of previous studies.

Most importantly, these cytoplasmic male sterile lines produced by protoplast fusion and interspecific hybridization usually showed poorer agronomic traits than maintainer lines, such as leaf yellowing, abnormal pistil, incomplete nectary development, seed pod deformity, flower bud yellowing, and abscission [46,47,48], but these undesirable traits were not observed in nuclear dominant male sterile lines without exogenous fragments [49]. Therefore, it is possible that these undesirable traits were caused by the disharmony between exogenous mitochondrial fragments and nuclear genes. Based on sequencing and comparative genome analysis, nine radish-specific fragments were identified in cabbage Ogura CMS with a total length of 35,618 bp, accounting for 13.84% of the mtDNA (Figure 4A). However, there were only four radish-specific fragments in Bel CMS with a total length of 21,587 bp, accounting for 8.93% of the mtDNA (Figure 6C). Ogura CMS is currently widely used in the production of cabbage hybrids, but it has the disadvantages of a high withered bud rate at the early flowering stage. Field phenotypic and pollen staining observations showed that Bel CMS had no withered buds (Figure 7), suggesting that the improvement of agronomic traits is likely to be related to the reduction of exogenous fragments. Moreover, those exogenous fragments may contain unknown genes or regulatory elements that can regulate the early development of flower buds. Cabbage Bel CMS is likely to replace Ogura CMS for cabbage heterosis utilization due to its less alien cytoplasm and better agronomic traits.

### 3.3. Comparison of Sterility Genes between Ogura CMS and Bel CMS in B. oleracea Plants

In homoplasmic male-sterile lines, sterility genes are mostly novel ORFs derived from mtDNA recombination and NHEJ, and they generally form chimeric structures together with known mitochondrial genes [27]. This feature has become one of the bases for identifying candidate genes for CMS. A variety of CMS genes have been identified by a comparative analysis of the mitochondrial genomes of cytoplasmic male-sterile lines and their maintainer lines, such as *orf113* in rice RT98A CMS [50], *orf507* in the pepper cytoplasmic male-sterile line FS4401 [51], *orf182* in rice D1-CMS [52], and *orf463a* in Radish NWB CMS [4]. In alloplasmic male-sterile lines, the CMS gene is derived from the donor parent. The candidate gene *orf346* in Nsa CMS *Brassica napus* has been demonstrated to be completely derived from the mitochondrial genome of *Arvensis* without any sequence variation [10,53]. Previous studies have shown that *orf138* in radish exhibits various polymorphisms, and they were classified into nine haplotypes (A~I) according to the difference in the nucleotide sequence [54]. The *orf138* gene in Ogura CMS cabbage line 19-3-2 belongs to haplotype A, which was consistent with the Ogura CMS cabbage line R2P2CMS [34].

Through restriction enzyme digestion and sequencing analysis, the sterility gene *orf138* was identified from *B. napus* cybrid 13 carrying Ogura CMS [55,56]. Subsequently, two homologous genes of *orf138* were discovered. The first was *orf125* in the Japanese cultivar Kozena [57] and Japanese wild radish [58], which is the sterility gene of Kos CMS. Compared to *orf138*, *orf125* had a 39-base deletion and two SNPs in the coding region (Figure 8). Kos CMS and its sterility gene, *orf125*, have been transferred into *B. napus* although protoplast fusion [59,60], and a 23-kb fragment derived from radish has been detected in the mitochondrial genome of cybrid progeny, accounting for 8.8% of mtDNA [16]. The other gene, *orf112*, was identified for the first time in the selfing progeny of *B. napus* cybrid 13F [61], with a deletion of 78 bases but no SNPs compared to *orf138*. However, there have been no reports about the use of this sterility gene. Sequence alignment showed that the 3′ ends of all three homologous genes had a 39-nucleotide sequence (AAAGGGGAAATAGAGGGGAAAGAGGAAAAAAAAGAGGGG), which was consecutively repeated three times in *orf138*, but there were only two repeats in *orf125* and only one repeat in *orf112* (Figure 8). At present, the function of this 39-bp repeat is unclear. The *orf112* gene we found in cabbage CMS line 19-2202 was identical to that in *B. napus*, and both of them were probably introduced from radish through protoplast fusion and intergeneric hybridization. According to the name of the original discoverer, the CMS line containing *orf112* was named Bel CMS to distinguish it from Ogura CMS and Kos CMS. We developed a molecular marker ‘OKB’ that can identify these three genes (OKB-F: TATTTTCTCGGTCCATTTTCCACCTCTT; OKB-R: GAATTTCAGTATGGGTGGCTAGGCGTC; annealing at 62 °C for 30 cycles), which can identify these three CMS types as well as fertile line efficiently and quickly at the seedling stage (Figure 8B). In addition, previous studies have shown that Ogura CMS and Kos CMS have the same restorer gene (*Rfo*/*Rfk*) [62], so the fertility of Bel CMS is likely to be restored by this restorer gene. We have hybridized Bel CMS with a fertile line containing the *Rfo* gene [1,63], and the fertility was successfully recovered, which showed that all three kinds of CMS can be fertility-restored by the same *Rfo* gene.

## 4. Materials and Methods

### 4.1. Plant Materials

The 305 cabbage materials used in this study were inbred lines and CMS lines, which were collected or created by the Institute of Vegetables and Flowers, Chinese Academy of Agricultural Sciences (IVF-CAAS). The original Ogura CMS materials were introduced from U.S.A. in 1998 by Prof. Zhiyuan Fang. 19-3-1 is a normal fertile cabbage inbred line. Different cabbage inbred lines were used as male parent to cross with donor CMS materials and successive backcross for multiple generations to generate different CMS cabbage lines, such as 19-3-2 and 19-2202. All the plant materials were grown in a greenhouse at IVF-CAAS.

### 4.2. Organelle Genome Sequencing, Assembly and Annotation

About 10 g of fresh cabbage leaves were harvested and quick-frozen in liquid nitrogen for DNA isolation using a modified extraction method. Genome sequencing of DNA samples was performed using second-generation Illumina HiSeq and third-generation ONT sequencing technology, after which an Illumina paired-end (PE) library (300–500 bp) and a Nanopore library (8–10 kb) were constructed. Sequencing was performed using NovaSeq 6000 and Nanopore high-throughput sequencing platforms.

The adapters, reads containing over 10% Ns (uncalled bases), duplicated sequences, and low-quality reads (the Phred scores < Q20) were removed. The Phred scores (Q20, Q30) and GC content of the clean data were calculated. SPADES 3.9.0 and Paired-Read Iterative Contig Extension (Price) software were used for genome assembly.

UGENE ORFS Finder was used to annotate the protein-coding genes, and tRNA annotations were submitted to the tRNAscan-se online website for annotation. RNAmmer 1.2 Server (https://services.healthtech.dtu.dk/service.php?RNAmmer-1.2/ (accessed on 2 December 2021)) was used to annotate the rRNAs. After the sequence annotation was finished, it was edited via Sequin to generate a submission file that could be submitted to the GenBank database. The edited GenBank annotation file was submitted to OGDRAW to construct an annotation map.

### 4.3. Collinearity Analysis and Sequence Alignment

The target and reference genomes were compared using MUMmer 3.23 software, and large-scale collinearity between the genomes was assessed. LASTZ 1.03.54 was then used to compare the regions to determine the local location arrangement relationships and identify Translocation/Trans, Inversion/Inv, and Translocation + Inversion (Trans + Inv) regions.

The online BLAST website (https://blast.ncbi.nlm.nih.gov/Blast.cgi (accessed on 2 December 2021)) was used for whole-genome sequence alignment. Small regions (<50 kb) and specific genes were compared using DNAMAN 6.0.3.99.

### 4.4. DNA Extraction and PCR Amplification

Total DNA was extracted from young leaves by the cetyl-trimethylammonium bromide (CTAB) method. All the primers used in this study were designed by Primer 5.0. PCR amplification was performed using a 20-μL reaction system, which included 7 μL of ddH_2_O, 10 μL of 2× Taq PCR Master Mix, 1 μL of DNA, and 1 μL of forward and reverse primers. The reaction procedure was as follows: predenaturation at 95 °C for 3 min; 35 cycles of denaturation at 95 °C for 15 s, annealing at 60 °C for 15 s and extension at 72 °C for 30 s; final extension at 72 °C for 5 min; and holding at 4 °C. The PCR products were subsequently detected via 1.5% agarose gel electrophoresis.

### 4.5. Pollen Staining Observation

At the early flowering stage, three plants were taken from each plot, and the top three wide-open flowers were taken from each plant. The anthers were collected and placed in a 2.0 mL centrifuge tube, supplemented with 200 μL Alexandria solution, and dyed at room temperature for 3–6 h. The stained anthers were stripped off to release pollen grains for microscopy observation. The red and rounded is the viable pollen, and the colorless and wrinkled is the unvital pollen.

## Figures and Tables

**Figure 1 ijms-22-13230-f001:**
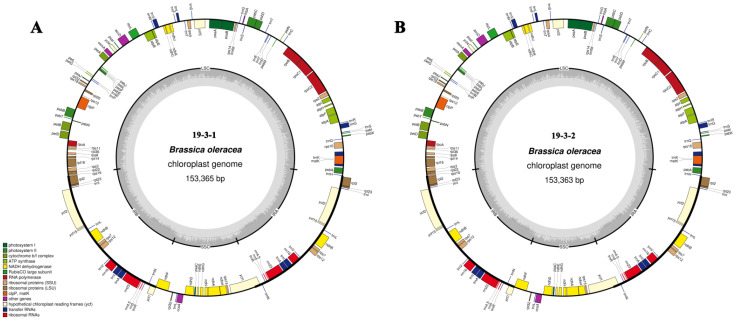
Chloroplast genome maps of normal-type (**A**) and Ogura-type (**B**) cabbage. The intraloop genes are shown with a clockwise transcription direction, while the extraloop genes are shown with the opposite. The different functional genes are shown in different colors. The gray histogram shows the GC content, and the gray line in the middle represents the 50% threshold line.

**Figure 2 ijms-22-13230-f002:**
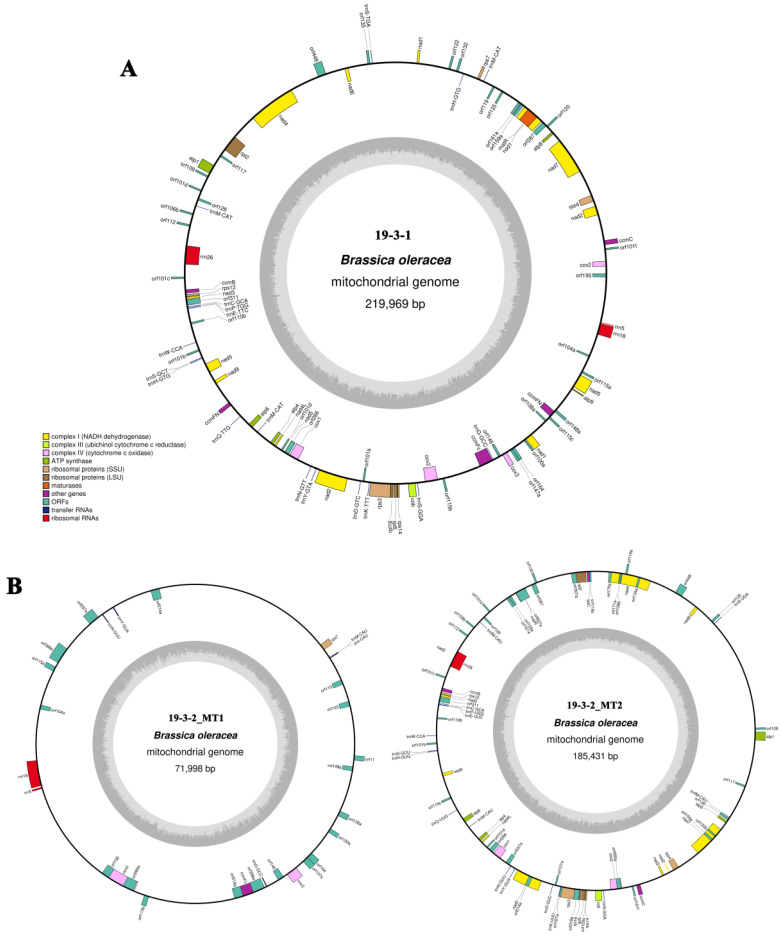
Mitochondrial genome maps of normal-type (**A**) and Ogura-type (**B**) cabbage. Genes with names inside the circle are transcribed clockwise. Genes with names outside the circle are transcribed counterclockwise. The colors of the genes denote the functions of the gene products.

**Figure 3 ijms-22-13230-f003:**
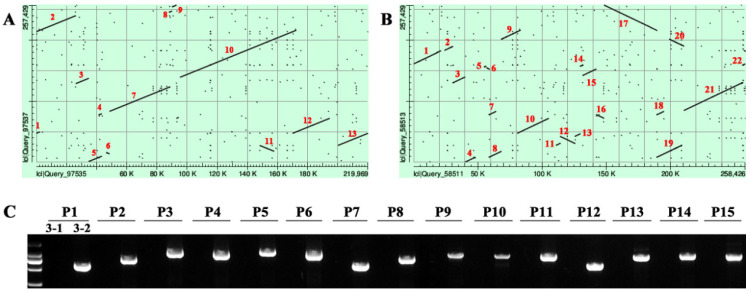
Synteny analysis of the mitochondrial genomes. (**A**) Normal-type cabbage on the *X*-axis, plotted against the Ogura-type cabbage on the *Y*-axis. (**B**) Ogura-type radish on the *X*-axis, plotted against the Ogura-type cabbage on the *Y*-axis. (**C**) Validation of structural variations between normal- and Ogura-type cabbage mtDNA. 3-1: normal-type cabbage 19-3-1; 3-2: Ogura-type cabbage 19-3-2; P1~P15: Primers 1~Primers 15.

**Figure 4 ijms-22-13230-f004:**
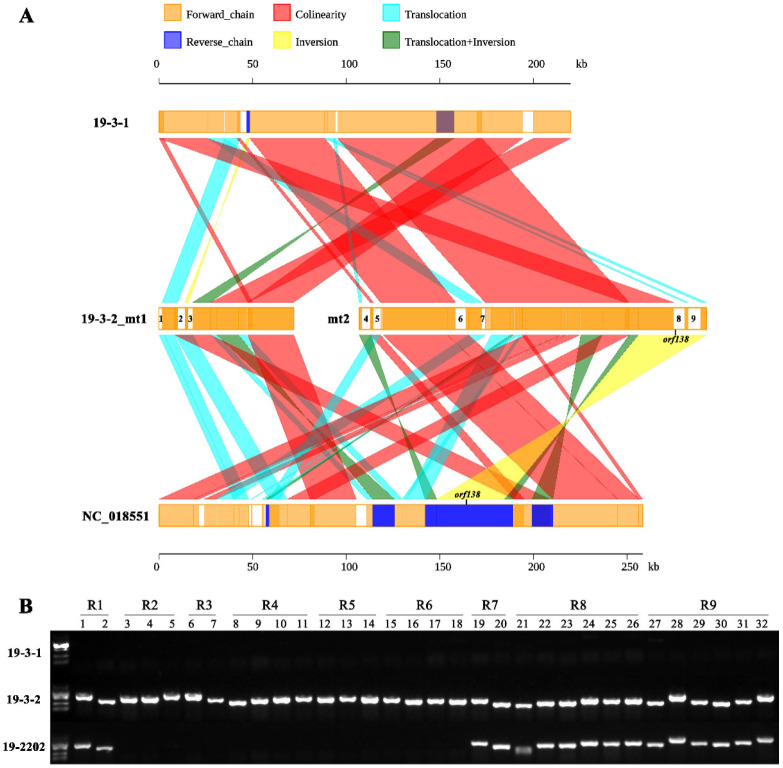
Distribution of alien cytoplasm in Ogura-type cabbage. (**A**) Syntenic comparison among 19-3-1, 19-3-2 and NC_018551 mtDNAs. (**B**) Detection results of alloplasmic markers. 19-3-1: Normal-type cabbage; 19-3-2: Ogura-type cabbage; NC_018551: Ogura-type radish; 19-2202: Bel CMS cabbage; R1~R9: Region 1~Region 9.

**Figure 5 ijms-22-13230-f005:**
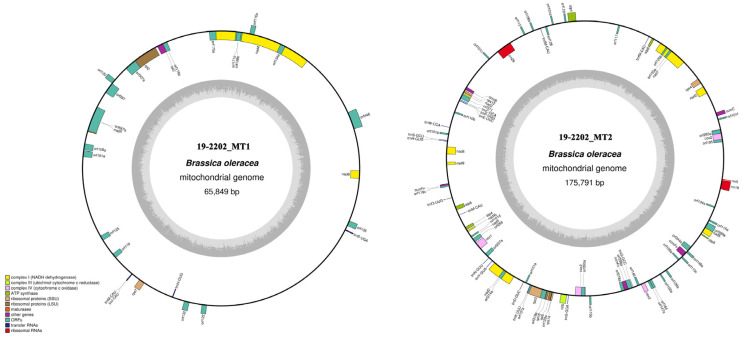
Mitochondrial genome map of the Bel CMS cabbage. Genes with names inside the circle are transcribed clockwise. Genes with names outside the circle are transcribed counterclockwise. The colors of the genes denote the functions of the gene products.

**Figure 6 ijms-22-13230-f006:**
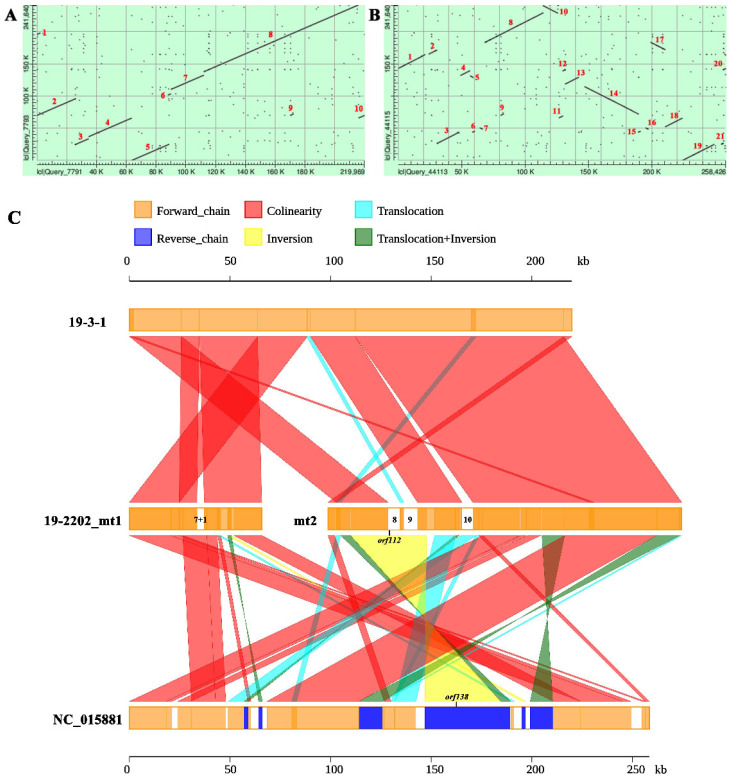
Syntenic comparison among 19-3-1, 19-2202 and NC_018551 mtDNAs. (**A**,**B**) Normal-type cabbage and Ogura-type radish on the *X*-axis, plotted against the Bel CMS cabbage on the *Y*-axis. (**C**) Distribution of alien cytoplasm in the Bel CMS mitochondrial genome. 19-3-1: Normal-type cabbage; 19-2202: Bel CMS cabbage. NC_018551: Ogura-type radish.

**Figure 7 ijms-22-13230-f007:**
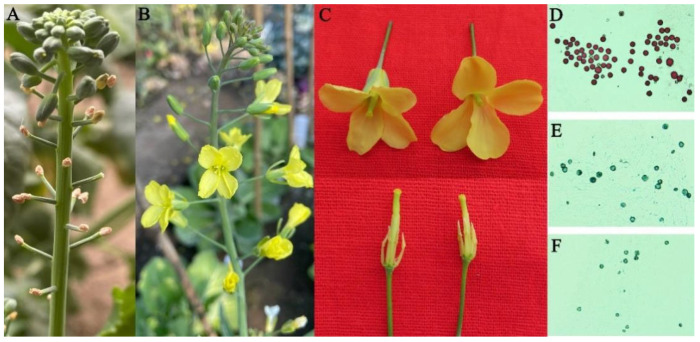
Phenotypic observations of Ogura CMS and Bel CMS cabbage under field conditions. (**A**,**B**) Flower buds of Ogura CMS (**A**) and Bel CMS (**B**) at the early flowering stage; (**C**) Floral organs between Ogura CMS (left) and Bel CMS (right); (**D**–**F**) Pollen staining observation in normal-type (**D**), Ogura CMS (**E**) and Bel CMS (**F**) cabbage.

**Figure 8 ijms-22-13230-f008:**
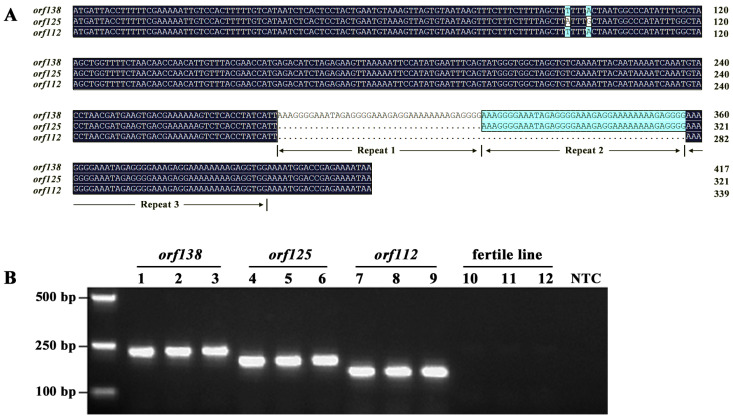
Difference of three homologous sterility genes. (**A**) Sequence alignment of three homologous sterility genes. (**B**) Amplification results of the molecular marker ‘OKB’. 1~3: Ogura CMS; 4~5: Kos CMS; 7~9: Bel CMS; 10~12: fertile line; NTC: no template control.

**Table 1 ijms-22-13230-t001:** Functional classification of chloroplast genomic protein-coding genes.

Function	Gene Name
ATP synthase	*atpA*, *atpB*, *atpE*, *atpF* *, *atpH*, *atpI*
Cytochrome b/f complex	*pet*A, *pe*tB, *pet*D, *pet*G, *pet*L, *pet*N
NADH dehydrogenase	*ndhA* *, *ndhB* *, *ndhC*, *ndhD*, *ndhE*, *ndhF*, *ndhG*, *ndhH*, *ndhI*, *ndh*J, *ndh*K
Photosystem I	*psaA*, *psaB*, *psaC*, *psaI*, *psaJ*
Photosystem II	*psbA*, *psbB*, *psbC*, *psbD*, *psbE*, *psbF*, *psbH*, *psbI*, *psbJ*, *psbK*, *psbL*, *psbM*, *psb*N, *psb*T, *psb*Z
Ribosomal protein (SSU)	*rps2*, *rps3*, *rps4*, *rps7*, *rps8*, *rps11*, *rps12*^#^, *rps14*, *rps15*, *rps16*, *rps*18, *rps*19
Ribosomal protein (LSU)	*rpl2* *, *rpl14*, *rpl16*, *rpl20*, *rpl22*, *rpl23*, *rpl32*, *rpl33*, *rpl*36
Ribosomal RNA	*rrn4.5*^1^, *rrn5*^1^, *rrn16*^1^, *rrn23*^1^
RNA polymerase	*rpoA*, *rpoB*, *rpoC1* *, *rpoC2*
Unknown function	*ycf1*^1^, *ycf2*, *ycf3* **, *ycf4*
Other gene	*accD*, *ccsA*, *cemA*, *clpP* **, *matK*, *rbcL*, *infA*

* Gene with one intron, ** Gene with two introns, ^#^ Transsplicing gene, ^1^ Gene with two copies in the IR region.

**Table 2 ijms-22-13230-t002:** Differences in coding genes among 19-3-2, 19-3-1 and NC_018551.

Gene	19-3-2 vs. 19-3-1		19-3-2 vs. NC_018551	
	Nucleotide	Amino Acid	Nucleotide	Amino Acid
*rrn18*	1398 A/G	Synonymous		
*atp8*			370 A/C	124 I/L
			449 T/C	150 V/A
*ccmB*			352 C/T	118 R/Q
*ccmC*			126 A/G	49 G/E
			146 G/A	Synonymous
			338 G/A	113 R/K
			351 G/A	Synonymous
			477 C/G	Synonymous
			533 G/C	178 G/A
			551 G/A	184 G/D
*ccmFC*			282 C/T	95 E/K
			305 G/T	102 F/L
			779 A/C	Synonymous
			1279 A/G	427 L/S
*ccmFN2*			377 T/C	126 L/S
*cox1*			108 G/A	Synonymous
			110 A/T	Synonymous
			799 A/C	Synonymous
*cox2*			378 A/G	127 W/R
*nad3*			264 A/G	Synonymous
*nad4*			77 C/T	26 P/L
			1205 C/T	402 P/L
			1399 G/C	467 V/L
*rpL16*			506 C/T	169 P/L
*rpL2*			464 G/A	155 G/D
			840 C/T	Synonymous
			919 G/A	Synonymous
			920 T/C	307 V/T
			1004 C/T	335 S/L
*rpL5*			515 T/C	172 L/P
*rps12*			12 A/C	4 F/L
			335 T/G	111 R/S
			344 T/G	Synonymous
*rps3*			75 T/G	25 D/E
			1254 A/C	Synonymous
			1320 A/C	Synonymous
*rps4*			188 A/C	103 S/F
			775 G/A	Synonymous

**Table 3 ijms-22-13230-t003:** Location and length of alien cytoplasm in Ogura-type cabbage mtDNA.

Region	Location	Start	End	Length	Number of Markers
R1	MT1	95	1645	1551	2
R2		9983	13,790	3808	3
R3		15,679	17,960	2282	2
R4	MT2	1501	5739	4239	4
R5		7573	11,390	3818	3
R6		51,431	56,887	5457	4
R7		65,521	67,247	1727	2
R8		167,906	173,821	5916	6
R9		175,701	182,520	6820	6

**Table 4 ijms-22-13230-t004:** Location and length of alien cytoplasm in the Bel CMS mitochondrial genome.

Region	Location	Start	End	Length
R7 + R1	MT1	33,649	37,217	3569
R8	MT2	29,996	35,831	5836
R9		37,715	44,534	6820
R10		66,628	71,989	5362

## Data Availability

The datasets used and/or analyzed during the current study are available from the corresponding author on reasonable request.

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
