# Peer review of "Organelle Comparative Genome Analysis Reveals Novel Alloplasmic Male Sterility with orf112 in Brassica oleracea L."

_ijms, 2021, doi:10.3390/ijms222413230_

Round 1

Reviewer 1 Report

Hybrid breeding is a dominant strategy of raising productivity of vegetable crops belonging to Brassicaceae family. Therefore, the research is highly relevant. The experimental data will be interesting both for breeders working with Brassica oleracea and geneticists studying organelle genomes. In the whole, the results are clearly presented but to improve

Comments

  1. It is necessary to substantiate the statement “Field phenotypic observation showed that Bel CMS was an excellent sterile source with stable 100% sterility and no withered buds at the early flowering stage, which could replace Ogura CMS in cabbage heterosis utilization.” (page 1). Any CMS type in any crop can be used in practical breeding when the reliable source of fertility restorer gene (or genes). A brief description of the fertility restoration genes suppressing CMS types considered is desirable.
  2. The origin of plant material with Bel CMS (line 19-2022) should be included in the section Material and methods. The authors focus the research novelity on the novel CMS type you should characterize the source material more accurate and widely. 

Reviewer 2 Report

To be able to understand this interesting paper, the authors should improve the methodology section, which is very week.

First, we have no idea about the number of samples used in this study.

Second, the description of the protocols are very brief, and they cannot allow understanding the massive data in the results section

Third, there is a big drawback around the statistics and their robustness. The authors should specify the number of samples and the criteria used to handle the data, including the bioinformatics. This is a prerequisite to understand the quality of the results the authors presented.

Fourth, the discussion is very weak. The authors are asked for further efforts to discuss first their results and then compare them to the large literature. What are the key home messages for each subsection/results. This should be clearly emphasized.

Firth, the figures captions, need much more work and descriptions. Please detail as much as possible to allow the reader to understand and read the graph without referring to the text. For example, the title/caption of Figure 5 is a nonsense. The figure 7 start by "comparison" by there is no statistical analysis. Be careful about any misunderstanding. Figure 8 is not clear and not informative at all, please remove.

Sixth, the abstract is not clear. Needs major revisions and modification. The context, the objectives, the methodology, the results and the conclusion. These all should be there.

Seventh, the conclusions should be further highlighted and discussed in terms of the hypothesis fixed in this work. Indeed, the hypothesis should be then clearly described in the introduction section.

Round 2

Reviewer 2 Report

The authors improved the manuscript.